# IMPROVING LLM FIRST-TOKEN PREDICTIONS IN MULTIPLE-CHOICE QUESTION ANSWERING VIA OUTPUT PREFILLING

## ABSTRACT

Large Language Models (LLMs) are traditionally evaluated on multiple-choice question answering (MCQA) tasks using *First-Token Probability* (FTP), which selects the answer option whose initial token has the highest likelihood. While efficient, FTP can be fragile: models may assign high probability to unrelated tokens (*misalignment*) or use a valid token merely as part of a generic preamble rather than as a clear answer choice (*misinterpretation*), undermining the reliability of symbolic evaluation. We propose a simple solution: *output prefilling*, a structured natural-language prefix (*e.g.*, 'The correct option is:') prepended to the model output. Originally explored in AI safety as an attack strategy, we repurpose prefilling to steer the model to respond with a clean, valid option, without modifying its parameters. Through extensive evaluation, we find that the FTP with prefilling strategy substantially improves accuracy, calibration, and output consistency across a broad set of LLMs and MCQA benchmarks. It outperforms standard FTP and often matches the performance of open-ended generation approaches that require full decoding and external classifiers, while being significantly more efficient. Our analysis suggests that prefilling is a simple, robust, and zero-cost method to enhance the reliability of FTP-based evaluation in multiple-choice settings.

## 1 INTRODUCTION

Large Language Models (LLMs) are increasingly deployed as general-purpose reasoning systems (Huang & Chang, 2023; Kojima et al., 2022; Plaat et al., 2024), where they are expected not only to generate fluent text, but also to make decisions (Liu et al., 2024; Lyu et al., 2025), answer questions (Liang et al., 2022; Kamalloo et al., 2023), and demonstrate understanding across a wide range of domains (Wei et al., 2022a; Naveed et al., 2023). A common way to evaluate these capabilities is through question-answering tasks, where the model is required to process a question and return a correct and relevant answer (Liang et al., 2022; Kamalloo et al., 2023; Chang et al., 2024; Khashabi et al., 2020). Among the many QA formats, one of the most widely adopted is **multiple-choice question answering (MCQA)**, in which the model selects from a fixed set of answer candidates, typically labeled A through D (Hendrycks et al., 2021; Lai et al., 2017; Clark et al., 2018; Li et al., 2024). MCQA benchmarks such as MMLU (Hendrycks et al., 2021) are widely used to evaluate the general knowledge, reasoning ability, and decision-making skills of a model across a broad range of subjects. In practice, models are typically prompted with a question and a set of answer options, and evaluation is performed by checking the model choice across the given, valid options.

There are various ways with differing tradeoffs to extract the model choice. Some approaches decode free-form generations into valid options either by using an auxiliary language model (*e.g.*, `GPT-3.5-Turbo` (Ouyang et al., 2022)) or a dedicated classifier (Yu et al., 2025; Wang et al., 2024a). While constraining outputs to a fixed format via an external model leads to more reliable evaluation, free-form generation alone may reduce alignment with human judgment (Molfese et al., 2025). Another approach, which does not require an external model, is to let the model generate one *single* token, and compare its probability across the *set of valid option tokens only* (*e.g.*, 'A', 'B', 'C', 'D'). This method, known as First-Token Probability (FTP) (Hendrycks et al., 2021; Santurkar et al., 2023), scores each candidate answer by computing the likelihood of it being generated as the first token and selecting the most probable one, without requiring full autoregressive decoding.

Figure 1: We show that a simple output prefilling template, which directs an LLM's first generated to-ken to a valid option for MCQA, substantially improves the standard first-token probability approach.

While FTP is efficient and model-agnostic, it relies on the assumption that the model is ready to commit to an answer immediately – an assumption that might break down in practice. In particular, models frequently diverge from the expected output structure, producing full sentences (*e.g.*, 'I believe the answer is A'), or ambiguous strings that are hard to interpret (*i.e.*, *first-token misalignment*). Worse still, we find that even when the first token is a valid label, it may serve a purely grammatical role that does not reflect the model intended answer (we call this *first-token misinterpretation*). For instance, 'A' could begin 'A possible answer could be C'. These ambiguities make first-token evaluation noisy and potentially misleading (Figure 1), eventually distorting accuracy metrics.

To make things worse, prior work has shown that MCQA model performance is highly sensitive to prompt design. Small variations in choice symbols, choice ordering, or phrasing can substantially affect model outputs, and thus their evaluation scores (Molfese et al., 2025; Balepur et al., 2025). This motivates why holding inputs fixed is needed for established benchmarks to ensure fair comparisons, and why relying on prompt design alone can not fully guarantee correct first-token outputs in zero-shot FTP evaluation, leaving a persistent risk of malformed or misaligned responses.

To address these limitations, in this work we introduce a simple yet effective solution by repurposing the *prefilling attack*, originally introduced in AI safety (Tang, 2024; Andriushchenko et al., 2025), to guide LLMs towards generating a valid option as the first token. Specifically, we build on the principle of *biasing model behavior through structured prefilling* and show that by prepending a short and benign phrase (such as 'The correct option is: ') effectively steers the model toward generating valid multiple-choice responses. Our experiments demonstrate that, despite its simplicity, this output prefilling approach substantially improves the reliability of FTP-based evaluation by reducing both the first-token misalignment and first-token misinterpretation issues, thus leading to consistent accuracy gains across different LLMs and benchmarks (*e.g.*, up to +40% on the Gemma-2-9B model (Team et al., 2024), when tested on the MMLU benchmark).

To further validate this, we conduct controlled experiments where the model selects its first token from the *full vocabulary*, rather than from the filtered valid options as in standard FTP, thereby allowing us to directly test whether the top-ranked token corresponds to the intended answer. Across models, prefilling consistently steers predictions toward the correct option, substantially reducing first-token misalignment – for instance, boosting Llama-3.1-8B's accuracy on SciQ (Welbl et al., 2017) from 2.2% to 96.9%. When compared against stronger baselines (*e.g.*, open-ended generation with GPT-based answer extraction), prefilling yields similarly aligned answers while being more efficient, as it removes the need for an auxiliary evaluation model. Beyond accuracy, we also assess its impact on model calibration (*i.e.*, the ability to represent predictive uncertainty). We find that across all models and configurations, calibration consistently improves by a substantial margin, demonstrating that our prefilling strategy enhances not only performance but also robustness.

Overall, our analysis shows output prefilling to be a simple zero-cost intervention that substantially improves the reliability of LLM outputs across diverse MCQA benchmarks. To our knowledge, this is the first work to rigorously quantify its effectiveness with modern general-purpose LLMs.

## 2 RELATED WORK

**LLMs as Classifiers.** Large Language Models (LLMs) have been increasingly used as zero-shot and few-shot classifiers by leveraging their in-context learning abilities (Brown et al., 2020; Chowdhery et al., 2023; Achiam et al., 2023). In multiple-choice question answering (MCQA), LLMs are typically prompted with a question followed by a list of labeled answer candidates. The model is then expected to produce or select the correct option, either via direct generation or scoring. Recent benchmarks such as MMLU (Hendrycks et al., 2021), RACE (Lai et al., 2017), and ARC (Clark et al., 2018) have become standard tools for assessing LLM performance across a range of domains.

To operationalize classification within a generative framework, these models are typically evaluated by measuring how likely they are to produce the correct label at the beginning of their output (Holtzman et al., 2021; Zhang et al., 2022; Min et al., 2022). This requires mapping free-form text generation into a constrained prediction task – often by prompting the model to respond with a symbolic label such as 'A' or 'B', corresponding to multiple-choice answers. Such symbolic setups allow for structured evaluation without retraining, but also introduce new sources of fragility tied to the surface-level generation behavior of the model (Wang et al., 2024a;b).

**First-Token Probability and Variants.** A common method for MCQA with LLMs is to decode the model output and evaluate whether the first generated token corresponds to the correct answer label (Zhang et al., 2022). This *first-token prediction* approach is appealing due to its simplicity and compatibility with generative models. However, it is also sensitive to decoding randomness, tokenization artifacts, and inconsistent formatting, often leading to ambiguous or incorrect outputs (Holtzman et al., 2021). Variants of this approach include ranking candidate completions by log-likelihood (Brown et al., 2020; Zhang et al., 2022), constraining decoding to answer tokens only (Zhao et al., 2021; Holtzman et al., 2021), or scoring options directly with logprob-based classification (Min et al., 2022; Yao et al., 2023).

**Prefilling Attack and Prompt Injection.** In the domain of AI safety, the *prefilling attack* (Tang, 2024; Andriushchenko et al., 2025) has emerged as a simple yet powerful prompt injection technique. It involves inserting innocuous-looking natural-language phrases into the model's output prompt to subvert safety filters or steer the model toward undesired behavior (Zou et al., 2023; Wei et al., 2023). For instance, adding a phrase such as 'Sure! The answer is:' can cause the model to comply with otherwise restricted user queries. These techniques expose vulnerabilities in even strongly aligned models, revealing the brittleness of instruction-following and safety mechanisms. While our work is not focused on bypassing safety constraints, we adopt the prefilling mechanism as a means of controlled behavioral biasing in MCQA, demonstrating its unintended but beneficial effects on model performance.

While techniques like chain-of-thought (Wei et al., 2022b), self-ask (Press et al., 2023), and tree-of-thoughts (Yao et al., 2023) involve guiding the model reasoning process through structured prompts, they do not guarantee strict adherence to a specific output format. In contrast, output-side prefilling directly inserts a fixed prefix into the model output, exploiting the normal cognitive biases of the model and thus ensuring that the first generated token aligns with the desired answer. This method enforces format consistency mechanically, which is particularly beneficial for tasks like FTP evaluation, where precise output formatting is crucial.

## 3 FIRST-TOKEN PROBABILITY AND OUTPUT PREFILLING

### 3.1 FIRST-TOKEN PROBABILITY

Let a general LLM be a function $f$ mapping a sequence of tokens $(t_1, t_2, \ldots, t_{N_t})$ drawn from a fixed vocabulary $\mathcal{V}$ into a distribution over next-token logits. These logits are typically normalized by a softmax function to obtain a probability distribution $P(t_{N_t+1})$ over the next token. Under greedy decoding, the token with the highest probability is selected as:

$$t_{N_t+1} = \arg\max_{t \in \mathcal{V}} P(t \mid t_1, t_2, \ldots, t_{N_t}). \tag{1}$$

**MCQA and First-Token Probability.** In MCQA, a model receives as input a question $q$ and a fixed set of symbolic answer options $\mathcal{A}^q = \{\text{'A'}, \text{'B'}, \text{'C'}, \text{'D'}\}$, and must select the most appropriate answer

Llama-3.1-8B:

```
<|begin_of_text|><|start_header_id|>user<|end_header_id|>
{QUESTION}
<|eot_id|><|start_header_id|>assistant<|end_header_id|>
Given the question and the possible options,
my answer is: {ANSWER}
```

Mistral-Nemo-12B:

```
<|im_start|>user<|im_sep|>{QUESTION}<|im_end|>
<|im_start|>assistant<|im_sep|>
Given the question and the possible options,
my answer is: {ANSWER}
```

Qwen-2-7B:

```
<|im_start|>user{QUESTION}<|im_end|>|im_start|>assistant
Given the question and the possible options,
my answer is: {ANSWER}
```

Phi-4-14B:

```
[INST]{QUESTION}[/INST]
Given the question and the possible options,
my answer is: {ANSWER}
```

Figure 2: Visual examples of our prefilling strategy. The prefilling template is added *after* the assistant's response start to mechanically condition the output format.

among them. To simplify evaluation, a common strategy is to restrict the probability distribution of the model to this small set of symbolic tokens, allowing for discrete scoring methods. FTP evaluates the model based solely on the probability it assigns to each answer token as the first word in its generated response. Given a formatted prompt (*e.g.*, 'Respond with the correct option: What is the capital of Italy? Answer:'), the model prediction is computed as

$$\arg\max_i P(a_i^q), \;\; a_i^q \in \mathcal{A}^q \tag{2}$$

where $a_i^q$ is the token representing the $i$-th answer choice and $P(a_i^q)$ is the probability assigned to it as the next token. While effective and efficient, this approach is fragile: the top-ranked token may lie outside the valid answer set (*i.e.*, misalignment), or the correct option might appear only after several tokens of open-ended generation (*i.e.*, misinterpretation). Together, these issues can lead to poor calibration, unreliable accuracy scores, and misleading model comparisons.

**First-Token Misalignment and Misinterpretation.** While the first-token probability approach is efficient and simple to implement, it suffers from two major failure modes. First, in *first-token misalignment*, the model may diverge from the expected symbolic output format, instead generating full sentences (*e.g.*, 'I believe the answer is A') or ambiguous strings that begin with irrelevant tokens. In such cases, the correct answer may be present later in the sequence, but the real, predicted token lies outside the valid answer set, making evaluation misleading. Second, even when the model first token *is* a valid label (*e.g.*, 'A'), it may serve a purely grammatical or stylistic function rather than indicating the intended answer – we call this *first-token misinterpretation*. For example, a response like 'A possible answer could be C' begins with a token in the answer set ('A') but ultimately points to a different choice ('C'), making the intended meaning unclear. These two phenomena introduce noise in model predictions, distort accuracy metrics, and hinder fair comparisons across prompting strategies and model architectures.

## 3.2 PREFILLING STRATEGY

**From AI Safety to MCQA.** To address the limitations of standard FTP evaluation, we propose an adaptation of the *prefilling attack*, a technique originally introduced in the AI safety community (Tang, 2024; Andriushchenko et al., 2025). In its adversarial form, the prefilling attack involves injecting a seemingly innocuous natural-language prefix (*e.g.*, 'Sure! Here's the answer:') to manipulate the model into generating otherwise restricted or unsafe content, effectively bypassing content filters. We repurpose this idea by prepending a short, benign prefix (*e.g.*, 'The correct option is:') to the model response prompt, to steer the model toward producing a valid symbolic answer as its very first token. This output prefilling intervention reduces ambiguity and misalignment by reinforcing the expected output format and encouraging direct, structured responses in MCQA tasks.

**Combining FTP and Prefilling.** Under our approach, the FTP method remains unchanged conceptually but is now applied after the assistant control token together with our prefilling prefix. This combined FTP–prefilling method preserves the efficiency and simplicity of FTP, while substantially mitigating issues of misalignment and misinterpretation, resulting in improved reliability, accuracy, and calibration across MCQA benchmarks.

In practice, to effectively inject these fixed tokens into the model output, we leverage the structured prompting conventions utilized by instruction-tuned LLMs. These models typically adhere to a specific dialogue format delineated by special control tokens (*e.g.*, '<|user|>' and '<|assistant|>'),

defining the response structure clearly. Although the exact formatting varies between models (for example, ChatML used in OpenAI models, Alpaca-style formatting for Llama derivatives, or ChatGLM native dialog structure), the prefilling prefix can always be placed immediately following the assistant starting token, effectively conditioning the subsequent generation of the model on the injected tokens.

Our key insight is that by inserting the prefilling prefix directly into the assistant output turn (Figure 2), the model treats this injected text as previously generated content, naturally continuing its response from the provided context and thus aligning closely with the expected MCQA format.

## 4 EXPERIMENTS

### 4.1 EXPERIMENTAL SETUP

**Benchmarks.** We evaluate our approach across several widely recognized MCQA benchmarks, each designed to test different cognitive capabilities and domains of knowledge. We begin with *general knowledge*, assessed using MMLU (Hendrycks et al., 2021), a comprehensive benchmark covering 57 academic and professional subjects. For *reading comprehension*, we employ RACE (Lai et al., 2017), a dataset derived from middle and high school exams and OpenBookQA (OBQA) (Mihaylov et al., 2018), which requires reasoning over scientific facts and general knowledge. In the domain of *commonsense reasoning*, we include Social IQa (SIQA) (Sap et al., 2019) for social interaction understanding, Moral Stories (MS) (Emelin et al., 2020) for moral decision-making and CommonsenseQA (CQA) (Talmor et al., 2019) for everyday commonsense reasoning. For *narrative understanding*, we consider Story Cloze (SC) (Mostafazadeh et al., 2016), where the model selects the most plausible story continuation, HellaSwag (HS) (Zellers et al., 2019), which evaluates narrative coherence in the presence of adversarial distractors, and MC-TACO (MC-T) (Zhou et al., 2019), which assesses temporal and causal reasoning in short texts. In the area of *STEM*, we use SciQ (Welbl et al., 2017) for scientific knowledge and MathQA (MQA) (Amini et al., 2019) to evaluate mathematical problem-solving. For *logical and analytical reasoning*, we consider AI2 Reasoning Challenge (ARC) (Clark et al., 2018), a large-scale benchmark of science exam questions. We use both the Easy partition (ARC-E), which mainly involves fact recall and can often be answered through direct retrieval, and the Challenge partition (ARC-C), which requires multi-step reasoning and the integration of scientific knowledge. We also include LogiQA (LQA) (Liu et al., 2020), based on national civil service exams. Together, these datasets enable a comprehensive evaluation of our method effectiveness and robustness across varied cognitive demands.

**Models.** We assess a range of open-source instruction-tuned LLMs, selected to cover diverse architectures, sizes, and alignment strategies. Our pool includes Meta's `Llama-3.1-8B`, Alibaba's `Qwen-2-7B`, DeepMind's `Gemma-7B` and `Gemma-2-9B`, HuggingFace's DPO-tuned `Zephyr-7B`, Microsoft's efficient `Phi-4-14B`, and two Mistral-family models: `Ministral-8B` and the alignment-focused `Mistral-Nemo-12B`. This selection enables a comprehensive analysis of prefilling effectiveness across different instruction-tuned models.

**Prompt Format.** For each benchmark, inputs are formatted as a concatenation of `general instruction`, `question`, and `answers`. To apply our prefilling strategy, we insert a natural-language prefix at the beginning of the model response. The default template used is: '`Given the question and the possible options, my answer is:`' that, placed immediately after the assistant message tag, promotes symbolic alignment and encourages structured, interpretable outputs. This template was selected based on preliminary trials, as it consistently yielded strong performance across all benchmarks and proved robust across different models. We consider as *valid answers* only those token sequences that match the symbolic answer labels exactly (*e.g.*, '`A`', '`B`', '`C`', '`D`'), optionally preceded by up to two whitespace or newline characters. A thorough analysis on the robustness of the proposed strategy to different templates are provided in the Appendix (cf. Table 5).

### 4.2 FTP VS. PREFILLING

We begin by evaluating the overall effectiveness of our prefilling strategy in enhancing the answer accuracy of our tested models across a diverse set of MCQA benchmarks, by comparing it with two natural baselines, the first one of which is the standard FTP approach. Specifically, for standard FTP, the model is prompted with the task and options, and the next-token probability distribution is used to score the first-token likelihoods assigned to each answer option. No prefix is injected. When using

Table 1: Benchmarking the fundamental abilities of base LLMs on natural language understanding, question answering, and reasoning tasks. Prefilling each model response positively impacts its ability to output the correct option, also against a direct prompt-instruction baseline. $\bar{\Delta}$ values denote average gains over base LLMs.

| | General | Comprehension | | Commonsense | | | Narrative | | | STEM | | Reasoning | | | |
|---|---|---|---|---|---|---|---|---|---|---|---|---|---|---|---|
| | MMLU | RACE | OBQA | SIQA | MS | CQA | SC | HS | MC-T | SciQ | MQA | ARC-E | ARC-C | LQA | $\bar{\Delta}$ |
| Llama-3.1-8B | 63.1 | 78.1 | 71.2 | 70.9 | 87.0 | 72.7 | 93.2 | 52.2 | 82.1 | 97.1 | 24.0 | 90.2 | 52.7 | 28.9 | |
| + prompting | 63.7 | 78.2 | 80.4 | 71.9 | 90.1 | **77.2** | 94.8 | 49.2 | **89.7** | 97.8 | 24.0 | 78.7 | 49.8 | 28.4 | |
| **+ prefilling** | **68.4** | **83.2** | **84.8** | **72.3** | **93.2** | 76.5 | **96.4** | **69.0** | 82.5 | **98.3** | **33.8** | **94.7** | **60.2** | **34.0** | **+6.0** |
| Qwen-2-7B | 68.9 | **87.0** | 83.8 | 75.6 | 88.1 | **79.4** | 97.1 | 53.4 | **88.3** | **97.5** | 29.8 | 94.3 | 48.0 | 33.0 | |
| + prompting | 68.3 | 86.7 | 84.2 | 75.2 | 85.3 | **79.4** | **97.7** | 65.0 | 85.1 | 97.2 | 29.8 | 74.1 | 47.9 | 31.6 | |
| **+ prefilling** | **69.1** | 86.7 | **84.8** | **76.1** | **92.6** | **79.4** | 97.3 | **65.2** | 86.1 | 97.4 | **36.5** | **94.7** | **56.6** | **35.5** | **+2.4** |
| Gemma-7B | 45.9 | 50.4 | 40.6 | 34.2 | **77.5** | 51.6 | 69.6 | 36.3 | 60.1 | 89.0 | 24.1 | 55.9 | 43.9 | 26.4 | |
| + prompting | 48.1 | 65.2 | 63.8 | 57.5 | 65.7 | 64.7 | 77.2 | 43.9 | **91.0** | 93.6 | 24.0 | 58.7 | 37.3 | **27.3** | |
| **+ prefilling** | **52.4** | **70.5** | **71.2** | **65.2** | 64.3 | **68.6** | **92.0** | **56.3** | **91.0** | **95.1** | **25.1** | **86.7** | **49.1** | 26.1 | **+14.9** |
| Gemma-2-9B | 33.9 | 49.5 | 38.8 | 56.3 | 85.9 | 39.6 | 97.6 | 43.0 | 72.2 | 31.8 | 27.8 | 38.6 | 52.4 | 31.0 | |
| + prompting | 70.2 | 86.2 | 87.6 | **74.2** | **91.6** | 78.7 | **97.9** | 63.6 | 77.1 | 98.1 | 27.8 | 81.2 | 55.9 | 32.1 | |
| **+ prefilling** | **72.1** | **86.3** | **89.8** | **74.2** | 87.3 | **79.0** | **97.9** | **69.8** | **78.5** | **98.3** | **34.0** | **96.6** | **60.8** | **35.8** | **+25.9** |
| Zephyr-7B | 57.5 | 86.5 | **73.0** | 67.5 | 83.6 | 66.0 | **96.0** | 35.0 | 69.1 | 88.5 | 23.3 | 95.6 | 51.0 | 32.1 | |
| + prompting | 55.9 | 72.3 | 67.6 | 66.5 | 85.3 | 54.4 | 91.7 | **40.0** | 85.2 | 89.6 | 23.3 | 65.7 | 41.8 | 30.1 | |
| **+ prefilling** | **58.8** | **87.3** | 70.8 | **68.5** | **86.2** | **69.0** | **96.0** | 36.1 | **93.3** | **93.1** | **24.3** | **98.1** | 51.0 | **40.6** | **+3.4** |
| Ministral-8B | 62.1 | **84.7** | 82.4 | 74.7 | 87.3 | 74.3 | 97.4 | 81.1 | 91.5 | 97.4 | 23.1 | 93.6 | 48.1 | 28.4 | |
| + prompting | 57.0 | 83.8 | 76.4 | 72.3 | 85.7 | 69.6 | 94.0 | 68.6 | 90.0 | 97.3 | 23.1 | 76.1 | 47.6 | 29.0 | |
| **+ prefilling** | **63.9** | **84.7** | **85.2** | **75.9** | **89.9** | **75.0** | **98.2** | **86.5** | **91.9** | **98.0** | **24.5** | **93.7** | **49.1** | **34.6** | **+1.8** |
| Mistral-Nemo-12B | 65.1 | 82.7 | 80.6 | 74.2 | 78.0 | 74.1 | **97.3** | 51.6 | 92.4 | 96.8 | 23.5 | 92.6 | 51.3 | 28.7 | |
| + prompting | 58.1 | 79.5 | 69.8 | 68.1 | 74.2 | 68.8 | 89.9 | 52.6 | 92.8 | 95.9 | 23.5 | 76.5 | 50.0 | 31.9 | |
| **+ prefilling** | **66.0** | **83.1** | **80.8** | **75.8** | **80.9** | **75.4** | 96.9 | **76.4** | **93.3** | **97.2** | **25.4** | **93.1** | **54.7** | **32.0** | **+3.0** |
| Phi-4-14B | 76.4 | 73.3 | 84.0 | 71.7 | 83.4 | 72.5 | 98.2 | 61.2 | 82.5 | 95.4 | 25.0 | **87.8** | 46.4 | 31.9 | |
| + prompting | 78.8 | **77.5** | **91.4** | 76.0 | 93.1 | **79.8** | **98.7** | 85.6 | 82.0 | 98.2 | 24.9 | 73.1 | 48.6 | 34.1 | |
| **+ prefilling** | **79.7** | 73.4 | 90.0 | **77.3** | **93.2** | 79.6 | **98.7** | **87.8** | **86.1** | **98.3** | **45.0** | **87.8** | **60.2** | **34.6** | **+7.3** |

our prefilling strategy, instead, a fixed natural-language prefix is injected at the beginning of each assistant response, before the model generates any output. The FTP scoring is then applied exactly as in the baseline. This approach retains the efficiency of FTP but adds a steering mechanism that helps the model align with the expected symbolic format, while adding no extra latency to the generation. In addition to the standard FTP, we include a prompt-engineering-based baseline, where the instruction 'Please answer only with [OPTIONS LIST]' is appended to the original prompt. This allows us to directly compare output-side prefilling with prompt-side guidance.

**Results.** Table 1 reports first-token accuracy across benchmarks, comparing performance of the base LLMs with prompt-side additions and with our output prefilling strategy. As shown, prefilling yields consistent improvements across all models and datasets. The largest average gain is observed for Gemma-2-9B, which improves by +25.9 points compared to standard FTP. Substantial improvements are also obtained for Gemma-7B (+14.9) and Zephyr-7B (+3.4). Even already strong models, such as Llama-3.1-8B and Phi-4-14B, achieve meaningful gains (+6.0 and +7.3, respectively), highlighting that prefilling is beneficial regardless of baseline capability.

In contrast, prompt-side additions lead to only modest or inconsistent improvements. Prefilling consistently outperforms prompt engineering, providing larger and more reliable accuracy gains. For example, Llama-3.1-8B improves by +10.4 points on ARC-C (60.2 vs. 49.8), Gemma-7B by +5.3 on RACE (70.5 vs. 65.2), Mistral-Nemo-12B by +23.8 on HS (76.4 vs. 52.6), and Phi-4-14B by +20.1 on MQA (45.0 vs. 24.9). These results demonstrate that mechanically enforcing the output format through prefilling is both more robust and broadly more effective than prompt-side instructions. These results confirm that structured prefilling reliably aligns the most probable model output with the intended symbolic answer, thereby reducing both misalignment and misinterpretation errors.

## 4.3 ALIGNMENT TO OPEN-ENDED GENERATION

To further contextualize our results, we compare our approach against an open-ended generation setting, which we consider a gold-standard reference due to its flexibility and alignment with natural model behavior – albeit at higher computational, training, maintenance, and evaluation cost due to decoding and external LLM classifier needs. In this setting, models generate free-form answers to each question without being constrained to select from symbolic options.

These open-ended responses are then mapped to symbolic answer labels through automatic evaluation using language model classifiers, such as `GPT-3.5-Turbo` (Ouyang et al., 2022), `Llama-3.1-70B-Instruct` (Dubey et al., 2024), and the `xFinder-Qwen` classifier (Yu et al., 2025), which is specifically designed and fine-tuned for robust and precise answer extraction from LLM outputs. The aim of this comparison is to measure how well our prefilling strategy approximates the answers that a model would naturally produce in an unconstrained, expressive scenario. As in previous settings, we use accuracy to quantify model performance by evaluating whether the extracted symbolic answer matches the ground-truth.

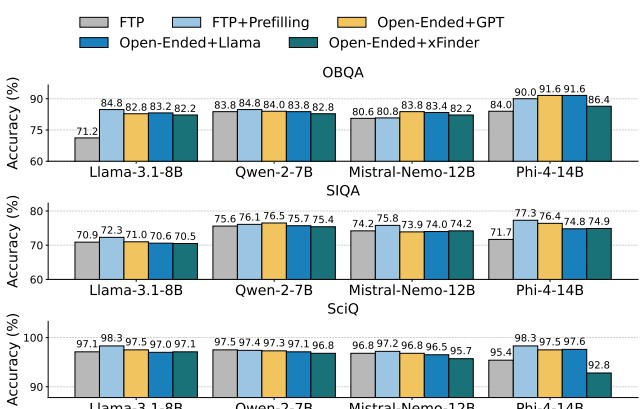

Figure 3: Comparison of model accuracy on OpenBookQA, Social IQa, and SciQ using standard FTP, FTP with prefilling, and open-ended generation with `GPT-3.5-Turbo`, `Llama-3.1-70B`, or `xFinder-Qwen` as classifiers. FTP with prefilling consistently outperforms standard FTP and often surpasses more expensive open-ended approaches.

**Results.** Figure 3 summarizes results on three representative benchmarks: OpenBookQA, Social IQa, and SciQ[1]. We observe that FTP with our prefilling strategy consistently achieves higher alignment with the symbolic labels derived from open-ended generation, outperforming the standard FTP baseline across all tasks. In several cases, FTP with prefilling even surpasses the accuracy of open-ended generation itself as judged GPT-based or xFinder-based evaluations, suggesting that it provides not only a more accurate but also a more efficient surrogate for expressive generation.

These results reinforce the idea that prefilling bridges the gap between efficient token-level scoring and natural free-form output, enabling structured symbolic accuracy without sacrificing alignment to the model unconstrained reasoning.

## 4.4 ANALYSIS AT FULL VOCABULARY

After assessing the effectiveness of prefilling, we bring more insights into the behavior of considered LLMs in an unconstrained decoding setting and we analyze what happens when we do not restrict scoring to a predefined set of symbolic answer tokens. Instead, we allow the model to generate freely from its entire vocabulary and record the top-1 predicted token. We then evaluate whether this token corresponds to a valid answer label (*e.g.*, 'A', 'B', 'C', 'D'), and whether it matches the correct answer for the given question. For this analysis, we report results using two metrics: the *full-vocabulary accuracy*, which captures how often the real top-1 token is both valid and correct without any filtering, and the First-Token Validity Rate (FTVR), which measures the proportion of examples where the top-1 token is a valid symbolic option.

Table 2: Full-vocabulary first-token evaluation on MMLU, OBQA, SIQA, and SCIQ. The model must predict a valid and correct token as its very first output. Prefilling improves both validity rate and accuracy.

| | MMLU | | OBQA | | SIQA | | SCIQ | |
|---|---|---|---|---|---|---|---|---|
| | Acc | FTVR | Acc | FTVR | Acc | FTVR | Acc | FTVR |
| `Llama-3.1-8B` | 6.4 | 9.5 | 17.8 | 20.4 | 52.7 | 69.6 | 2.2 | 2.4 |
| **+ prefilling** | **64.0** | **99.3** | **80.8** | **99.8** | **71.5** | **99.9** | **96.9** | **100.0** |
| `Qwen-2-7B` | 61.7 | 85.9 | 80.4 | 93.8 | 71.6 | 90.9 | 94.3 | 95.7 |
| **+ prefilling** | **66.1** | **97.0** | **84.8** | **100.0** | **75.8** | **99.4** | **98.0** | **99.9** |
| `Mistral-Nemo-12B` | 21.6 | 27.6 | 31.4 | 34.6 | **44.5** | **56.5** | 3.2 | 3.5 |
| **+ prefilling** | **40.7** | **61.9** | **64.8** | **77.2** | 42.8 | 53.9 | **72.0** | **73.1** |
| `Phi-4-14B` | 8.9 | 10.7 | 36.8 | 42.6 | 30.3 | 35.7 | 10.1 | 11.7 |
| **+ prefilling** | **37.0** | **41.2** | **81.0** | **88.0** | **72.7** | **93.3** | **85.4** | **86.7** |

As shown in Table 2 and Table 6 of the Appendix, applying prefilling substantially improves both FTVR and accuracy across all models.

---

[1]The complete results on all benchmarks are reported in Appendix B, as well as the exact prompting and classification details for open-ended responses.

Table 3: Calibration results on four MCQA benchmarks in terms of Adaptive Calibration Error (ACE), Brier score, and Log Loss, before and after applying prefilling. Prefilling consistently achieves lower values across all metrics, indicating improved alignment between model confidence and actual correctness. For all metrics, lower the better (↓).

|  | MMLU | | | OBQA | | | SIQA | | | SCIQ | | |
|---|---|---|---|---|---|---|---|---|---|---|---|---|
|  | ACE | Brier-S | LogLoss | ACE | Brier-S | LogLoss | ACE | Brier-S | LogLoss | ACE | Brier-S | LogLoss |
| Llama-3.1-8B | 0.206 | 23.4 | 0.815 | 0.444 | 47.5 | 1.987 | 0.169 | 28.4 | 0.795 | 0.007 | 2.2 | 0.077 |
| + prefilling | **0.129** | **18.7** | **0.616** | **0.252** | **45.5** | **1.341** | **0.138** | **19.2** | **0.752** | **0.006** | **1.7** | **0.048** |
| Qwen-2-7B | 0.246 | 25.2 | 1.311 | 0.327 | 50.8 | 1.790 | 0.210 | 21.8 | 1.213 | 0.020 | 2.1 | 0.112 |
| + prefilling | **0.213** | **23.0** | **0.894** | **0.289** | **46.3** | **1.495** | **0.190** | **19.8** | **1.126** | **0.014** | **1.8** | **0.096** |
| Mistral-Nemo-12B | 0.173 | 21.3 | 0.673 | 0.288 | **45.9** | 1.414 | 0.157 | 20.6 | 0.707 | 0.007 | 2.3 | 0.079 |
| + prefilling | **0.164** | **20.4** | **0.624** | **0.252** | 46.3 | **1.358** | **0.115** | **18.5** | **0.600** | **0.004** | **1.9** | **0.064** |
| Phi-4-14B | 0.187 | **36.4** | 1.084 | 0.340 | 50.3 | 1.712 | 0.210 | 24.1 | 1.547 | 0.034 | 3.7 | 0.177 |
| + prefilling | **0.167** | 36.6 | **1.044** | **0.272** | **47.4** | **1.421** | **0.191** | **19.6** | **1.438** | **0.013** | **1.4** | **0.088** |

Notably, models such as Gemma-2-9B and Llama-3.1-8B experience gains of over 60 points. These results confirm that prefilling shifts the full next-token distribution toward structured symbolic output, making first-token decoding a more reliable proxy for answer selection in MCQA tasks.

## 4.5 Calibration Analysis

To further motivate our prefilling strategy, we evaluate the calibration of model predictions, focusing on how well the model confidence scores align with their actual accuracy. A well-calibrated model is one whose predicted confidence closely reflects the true likelihood of correctness. For example, predictions made with 80% confidence should be correct roughly 80% of the time. When this relationship breaks down, it indicates miscalibration: overconfidence occurs when confidence exceeds accuracy, while underconfidence reflects the opposite. Improving calibration is critical for increasing the reliability of model outputs, particularly in high-stakes or trust-sensitive applications.

To quantify calibration, we use three metrics: Adaptive Calibration Error (ACE) (Nixon et al., 2019), Brier score (Brier-S) (Glenn et al., 1950), and Log Loss (Hastie et al., 2009; LeCun et al., 2015). ACE improves upon traditional metrics like Expected Calibration Error (ECE) (Naeini et al., 2015) by addressing issues related to fixed binning and multi-class settings. Lower ACE values indicate better calibration. The Brier-S, on the other hand, captures the mean squared difference between predicted probabilities and binary outcomes. A lower Brier-S indicates both accurate and well-calibrated predictions, while a higher score penalizes incorrect predictions made with high certainty. Log Loss, instead, measures the discrepancy between predicted probabilities and true labels across all classes. It is particularly sensitive to overconfident incorrect predictions, complementing the Brier-S by emphasizing mistakes made with high certainty. The formal definitions of each metric are provided in Appendix D.

The results are reported in Table 3, showing that prefilling consistently improves model calibration. In particular, examining the ACE metric, Llama-3.1-8B improves substantially on MMLU (from 0.206 to 0.129) and OpenBookQA (from 0.444 to 0.252). Even stronger baselines such as Phi-4-14B benefit from prefilling, with ACE on OpenBookQA dropping from 0.340 to 0.272. Consistent gains are also observed in terms of Brier-S, confirming that models become not only more accurate but also less overconfident in their incorrect predictions. This is especially noticeable on Social IQa, where overconfident errors are common: for instance, Llama-3.1-8B improves from 28.4 to 19.2, and Phi-4-14B from 24.1 to 19.6. Turning to Log Loss, prefilling reduces scores across all benchmarks and models. For example, Llama-3.1-8B decreases from 1.987 to 1.341 on OpenBookQA, and Phi-4-14B from 1.712 to 1.421 on the same benchmark. Since Log Loss heavily penalizes overconfident errors, these reductions indicate that prefilling helps models produce probability estimates that are better calibrated and more reliable.

The calibration diagrams in Figure 4 provide a more detailed view across different confidence levels. We observe that models such as Gemma-2-9B and Zephyr-7B tend to be overconfident in the mid-confidence range, while Llama-3.1-8B remains overconfident even at maximum predicted confidence (100%). This calibration analysis suggests that, in addition to improving accuracy, prefilling also enhances the trustworthiness of model confidence estimates.

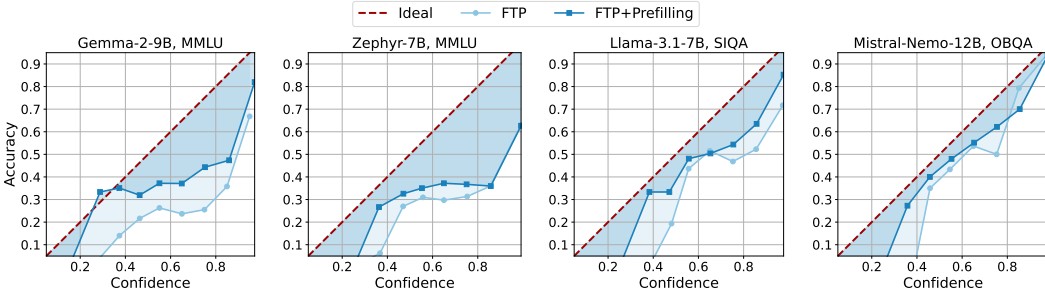

Figure 4: Calibration curves comparing standard FTP and FTP with prefilling. Prefilling improves calibration, moving predictions closer to the ideal confidence (*i.e.*, accuracy alignment).

## 4.6 SYMBOLIC OUTPUT STRUCTURE ANALYSIS

While standard accuracy metrics quantify how often a model fails, they offer limited insight into how models fail. To deepen our understanding of model behavior, we analyze auxiliary statistics that capture qualitative patterns in error generation.

Specifically, to analyze robustness against first-token misinterpretation, we introduce a new metric that answers the following question: *When the model predicts a valid symbolic answer (*e.g., 'A', 'B'),* how consistent is the continuation?* Ideally, an aligned model should either stop after the symbolic token or follow it with a predictable structure. In contrast, inconsistent or verbose continuations may indicate that the answer token was generated as part of a grammatical phrase rather than as an intended choice. We capture this behavior using the Continuation Diversity (CD), defined as the number of distinct second tokens that follow a valid first-token prediction, in relation to the calculated First-Token Validity Rate (FTVR). Formally, it is computed as $CD = \frac{S}{FTVR}$, where $S$ is the number of unique second tokens that follow a valid first-token prediction. Low CD suggests that the

Table 4: Evaluation of symbolic output structure across four MCQA benchmarks in terms of CD and FTVR. Prefilling consistently increases FTVR and lowers CD, indicating that it reduces misinterpretation errors and promotes more stable symbolic completions.

|  | MMLU | | OBQA | | SCIQ | |
|---|---|---|---|---|---|---|
|  | CD (↓) | FTVR (↑) | CD (↓) | FTVR (↑) | CD (↓) | FTVR (↑) |
| Llama-3.1-8B | 4.38 | 9.5 | 0.44 | 20.4 | 0.42 | 2.4 |
| **+ prefilling** | **0.05** | **99.3** | **0.02** | **99.8** | **0.01** | **100.0** |
| Qwen-2-7B | 0.30 | 85.9 | 0.06 | 93.8 | 0.03 | 95.7 |
| **+ prefilling** | **0.04** | **97.0** | **0.03** | **100.0** | **0.01** | **99.9** |
| Mistral-Nemo-12B | 0.19 | 27.6 | 0.02 | 34.6 | 0.11 | 3.5 |
| **+ prefilling** | **0.09** | **61.9** | **0.01** | **77.2** | **0.01** | **73.1** |
| Phi-4-14B | 7.86 | 10.7 | 0.90 | 42.6 | 5.14 | 11.7 |
| **+ prefilling** | **0.10** | **41.2** | **0.01** | **88.0** | **0.01** | **86.7** |

model reliably adheres to a symbolic format (*e.g.*, consistently producing 'A.'), whereas high CD may signal poor format consistency and potential misinterpretation.

As shown in Table 4, the prefilling strategy consistently improves this metric by substantially reducing the diversity of second-token continuations. Notably, values approaching or surpassing 1 indicate that the number of distinct second tokens is comparable to or larger than the number of valid first-token predictions, suggesting unstable or inconsistent generation patterns.

## 5 CONCLUSION

This work demonstrates the significant benefits of using output prefilling to enhance the reliability of FTP evaluation in MCQA tasks. By adding a structured prefix to the model response template, we substantially improve issues of first-token misalignment and misinterpretation, steering the model toward more accurate and consistent predictions. Experiments across several MCQA benchmarks show that this method not only improves alignment between the predicted token and the correct answer but also boosts overall accuracy and calibration, with FTP accuracies comparable to those obtained via open-ended generation evaluated by external GPT, Llama, or xFinder classifiers. Notably, these gains are achieved without fine-tuning or any model modifications. To our knowledge, this is the first work to rigorously quantify the effectiveness of prefilling with modern general-purpose LLMs, highlighting its promise as a lightweight strategy for choice-based evaluation scenarios.

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

## A  ANALYSIS ACROSS MULTIPLE TEMPLATES

In addition to the main experiments, which utilize the default prefilling template (*i.e.*, `Given the question and the possible options, my answer is: `), we conduct a supplementary analysis to assess the robustness of models performance across template variations. Specifically, we evaluate the models using a set of 10 distinct templates, each phrased differently but conveying the same task. For this analysis, we report the average accuracy and standard deviation across the 10 templates to capture both overall performance and sensitivity to prompt formulation.

The set of templates used in this analysis alternates short, concise wording to longer, more context-aware versions, like:

- `I choose: `
- `Having evaluated the question and its choices, I conclude with: `
- `My final answer is: `
- `Upon careful reflection, the response I find most appropriate is: `
- `Alright, I'm going with: `
- `After reviewing the options thoughtfully, I've decided on: `
- `Given the question and the possible options, my answer is: `
- `Let's cut to the chase, the answer is: `
- `After thorough consideration of the question and all potential answers, my final selection is: `
- `Given the context and underlying assumptions in both the question and its options, I determine the most fitting response to be: `

The average accuracies, together with their standard deviations, are reported in Table 5, alongside the results already displayed in Table 1. From the results, we can observe that these average accuracies are generally comparable to those obtained using the single default template. These results demonstrate that the prefilling strategy is robust across different template formulations and is not highly sensitive to template phrasing. In a few rare cases, however, significant improvements are observed. For example, `Gemma-7B` achieves an accuracy 5.9 percentage points higher than the default template on the Moral Stories benchmark, while `Phi-4-14B` shows a 5.8-point gain on the LogiQA benchmark. These results suggest that the default template may not always be the most effective choice.

This variability aligns with the understanding that no single template style is optimal across all models. Differences in architecture, pretraining corpora, and alignment strategies influence how models interpret and get affected by certain tokens. As a result, a template that benefits one model may be suboptimal for another.

Moreover, we find that longer templates can exert a stronger influence on the model generation. While this added guidance is generally beneficial, leading to better task alignment, it may also introduce unintended biases or dilute the task signal in others. These variations open opportunities for users to tailor template styles to the specific model they are using, while retaining confidence that reasonable rephrasings do not significantly degrade performance.

## B  OPEN-ENDED GENERATION

**Prompting and Classification Setup.** We leverage `GPT-3.5-Turbo`, `Llama-3.1-70B-Instruct`, and `xFinder-Qwen` to classify the open-ended generations produced by the models in response to benchmark questions. To generate these responses, we follow the same prompting procedure used for the FTP method: the model is presented with the question and all possible answer options, and in this case, it is allowed to generate freely until the EOS token.

To convert each open-ended response into a symbolic answer label, we prompt `GPT-3.5-Turbo`, `Llama-3.1-70B-Instruct`, and `xFinder-Qwen` with the following instruction:

```
Given these possible options:
{options}\n
And this given output:
{response}\n
Classify the output into one and only one
of the aforementioned options.
```

Table 5: Average accuracies and standard deviations across 10 distinct prompt templates using prefilling. Results are compared against those obtained with the default template and without prefilling. This highlights the robustness of the prefilling strategy to prompt rephrasing.

| | General | Comprehension | | Commonsense | | | Narrative | | | STEM | | Reasoning | | |
|---|---|---|---|---|---|---|---|---|---|---|---|---|---|---|
| | MMLU | RACE | OBQA | SIQA | MS | CQA | SC | HS | MC-T | SciQ | MQA | ARC-E | ARC-C | LQA |
| `Llama-3.1-8B` | 63.1 | 78.1 | 71.2 | 70.9 | 87.0 | 72.7 | 93.2 | 52.2 | 82.1 | 97.1 | 24.0 | 90.2 | 52.7 | 28.9 |
| **+ prefilling** (default) | **68.4** | **83.2** | **84.8** | **72.3** | **93.2** | **76.5** | **96.4** | **69.0** | **82.5** | **98.3** | 33.8 | **94.7** | **60.2** | **34.0** |
| **+ prefilling** (avg) | 66.9 | 82.9 | 81.7 | 71.8 | 91.3 | 76.6 | **96.4** | 66.0 | 79.0 | 98.1 | **34.9** | 92.7 | 58.8 | 33.3 |
| | ±8.6e⁻³ | ±5.9e⁻³ | ±1.4e⁻² | ±6.6e⁻³ | ±2.8e⁻² | ±5.6e⁻³ | ±7.7e⁻³ | ±2.8e⁻² | ±5.5e⁻² | ±1.4e⁻² | ±9.6e⁻³ | ±3.9e⁻³ | ±9.0e⁻³ | ±6.2e⁻³ |
| `Qwen-2-7B` | 68.9 | **87.0** | 83.8 | 75.6 | 88.1 | 79.4 | 97.1 | 53.4 | **88.3** | 97.5 | 29.8 | 94.3 | 48.0 | 33.0 |
| **+ prefilling** (default) | **69.1** | 86.7 | **84.8** | **76.1** | **92.6** | 79.4 | 97.3 | 65.2 | 86.1 | 97.4 | 36.5 | **94.7** | **56.6** | **35.5** |
| **+ prefilling** (avg) | 68.9 | 86.6 | 84.4 | **76.1** | 91.7 | **79.5** | 97.5 | 65.3 | 86.7 | 97.5 | **37.3** | **94.7** | 55.9 | 33.1 |
| | ±8.6e⁻³ | ±3.4e⁻³ | ±1.0e⁻² | ±6.0e⁻³ | ±4.8e⁻² | ±6.1e⁻³ | ±2.6e⁻³ | ±6.0e⁻² | ±4.5e⁻² | ±3.7e⁻³ | ±1.4e⁻² | ±1.9e⁻³ | ±2.0e⁻² | ±6.2e⁻³ |
| `Gemma-7B` | 45.9 | 50.4 | 40.6 | 34.2 | **77.5** | 51.6 | 69.6 | 36.3 | 60.1 | 89.0 | 24.1 | 55.9 | 43.9 | **26.4** |
| **+ prefilling** (default) | 52.4 | **70.5** | **71.2** | 65.2 | 64.3 | **68.6** | 92.0 | **56.3** | **91.0** | **95.1** | 25.1 | 86.7 | 49.1 | 26.1 |
| **+ prefilling** (avg) | **52.7** | 70.3 | 70.9 | **65.3** | 70.4 | 68.3 | **92.1** | 55.5 | 89.6 | 94.9 | **25.7** | **86.8** | **49.6** | 26.2 |
| | ±3.8e⁻³ | ±2.5e⁻³ | ±7.2e⁻³ | ±4.5e⁻³ | ±7.1e⁻² | ±4.7e⁻³ | ±6.5e⁻³ | ±2.4e⁻² | ±2.3e⁻² | ±2.6e⁻³ | ±1.2e⁻² | ±3.5e⁻³ | ±1.5e⁻² | ±6.5e⁻³ |
| `Gemma-2-9B` | 33.9 | 49.5 | 38.8 | 56.3 | 85.9 | 39.6 | 97.6 | 43.0 | 72.2 | 31.8 | 27.8 | 38.6 | 52.4 | 31.0 |
| **+ prefilling** (default) | **72.1** | 86.3 | **89.8** | 74.2 | 87.3 | **79.0** | **97.9** | 69.8 | **78.5** | **98.3** | 34.0 | **96.6** | **60.8** | **35.8** |
| **+ prefilling** (avg) | 71.6 | **86.4** | 89.4 | **74.4** | **88.0** | 78.9 | 97.8 | **70.9** | 77.3 | **98.3** | **34.5** | **96.6** | 59.8 | 34.1 |
| | ±3.5e⁻³ | ±3.3e⁻³ | ±1.1e⁻² | ±2.1e⁻³ | ±4.1e⁻² | ±4.7e⁻³ | ±1.9e⁻³ | ±5.2e⁻² | ±2.1e⁻² | ±1.1e⁻³ | ±2.0e⁻² | ±1.0e⁻³ | ±2.0e⁻² | ±1.1e⁻² |
| `Zephyr-7B` | 57.5 | 86.5 | **73.0** | 67.5 | 83.6 | 66.0 | 96.0 | 35.0 | 69.1 | 88.5 | 23.3 | 95.6 | 51.0 | 32.1 |
| **+ prefilling** (default) | **58.8** | **87.3** | 70.8 | **68.5** | **86.2** | **69.0** | 96.0 | 36.1 | **93.3** | 93.1 | 24.3 | **98.1** | 51.0 | **40.6** |
| **+ prefilling** (avg) | 58.1 | 73.4 | 71.6 | 67.5 | 85.7 | 68.5 | **96.3** | **36.3** | 91.0 | 93.8 | **24.7** | 87.8 | **52.2** | 33.1 |
| | ±2.1e⁻³ | ±2.6e⁻³ | ±1.1e⁻² | ±6.8e⁻³ | ±3.9e⁻² | ±1.4e⁻² | ±3.9e⁻³ | ±2.0e⁻² | ±4.2e⁻² | ±6.5e⁻³ | ±1.7e⁻² | ±3.5e⁻³ | ±1.4e⁻² | ±1.3e⁻² |
| `Ministral-8B` | 62.1 | **84.7** | 82.4 | 74.7 | 87.3 | 74.3 | 97.4 | 81.1 | 91.5 | 97.4 | 23.1 | 93.6 | 48.1 | 28.4 |
| **+ prefilling** (default) | **63.9** | **84.7** | **85.2** | **75.9** | 89.9 | 75.0 | **98.2** | **86.5** | **91.9** | **98.0** | **24.5** | **93.7** | 49.1 | **34.6** |
| **+ prefilling** (avg) | 63.5 | 84.3 | 84.2 | 75.5 | **90.0** | **75.3** | 98.0 | 85.9 | 91.6 | 97.6 | **24.5** | 93.4 | **51.3** | 32.2 |
| | ±4.2e⁻³ | ±6.5e⁻³ | ±1.2e⁻² | ±5.4e⁻² | ±3.1e⁻² | ±4.4e⁻³ | ±1.6e⁻² | ±1.0e⁻² | ±2.0e⁻² | ±2.6e⁻³ | ±1.4e⁻² | ±2.7e⁻³ | ±3.3e⁻² | ±1.1e⁻² |
| `Mistral-Nemo-12B` | 65.1 | 82.7 | 80.6 | 74.2 | 78.0 | 74.1 | **97.3** | 51.6 | 92.4 | 96.8 | 23.5 | 92.6 | 51.3 | 28.7 |
| **+ prefilling** (default) | **66.0** | **83.1** | 80.8 | **75.8** | 80.9 | **75.4** | 96.9 | **76.4** | **93.3** | **97.2** | 25.4 | **93.1** | **54.7** | **32.0** |
| **+ prefilling** (avg) | 65.0 | 83.0 | 79.8 | 75.1 | **82.4** | 74.3 | 97.0 | 73.1 | 91.7 | **97.2** | **25.7** | 92.5 | 54.4 | 31.4 |
| | ±7.4e⁻³ | ±1.0e⁻² | ±1.2e⁻² | ±4.6e⁻³ | ±9.0e⁻² | ±1.1e⁻² | ±3.4e⁻³ | ±3.4e⁻² | ±3.2e⁻² | ±4.0e⁻³ | ±1.5e⁻² | ±7.9e⁻³ | ±2.1e⁻² | ±6.9e⁻³ |
| `Phi-4-14B` | 76.4 | 73.3 | 84.0 | 71.7 | 83.4 | 72.5 | 98.2 | 61.2 | 82.5 | 95.4 | 25.0 | 87.8 | 46.4 | 31.9 |
| **+ prefilling** (default) | **79.7** | 73.4 | **90.0** | **77.3** | 93.2 | **79.6** | **98.7** | **87.8** | **86.1** | **98.3** | 45.0 | 87.8 | **60.2** | 34.6 |
| **+ prefilling** (avg) | 79.0 | **87.2** | 89.6 | 76.5 | **94.1** | 79.4 | 98.4 | 86.7 | 81.2 | 98.0 | **45.2** | **98.1** | 59.6 | **40.4** |
| | ±2.1e⁻³ | ±2.7e⁻³ | ±1.4e⁻² | ±1.1e⁻² | ±3.7e⁻² | ±6.2e⁻³ | ±3.5e⁻³ | ±1.7e⁻² | ±1.0e⁻¹ | ±4.4e⁻³ | ±2.8e⁻² | ±2.3e⁻³ | ±3.3e⁻² | ±2.1e⁻² |

```
Return only the option letter
(A, B, C, etc.).
```

where `options` contains the list of candidate answers in the format:

```
A) option_1
B) option_2
...
```

and `response` is the text generated by the evaluated model. In all cases, the predicted classification is consistently returned either as the bare letter (*e.g.*, 'B') or the letter followed by parenthesis (*e.g.*, 'B) '). We then compare the predicted letter with the ground-truth answer to compute the final accuracy.

**Expanded Comparison with Llama- and xFinder-Based Classification.** In addition to the analysis presented in Section 4.3, we evaluate the open-ended generations produced by all models across the benchmark datasets. The results using `Llama-3.1-70B-Instruct` are reported in Figure 5, while the results using the `xFinder-Qwen` classifier are reported in Figure 6. `Llama-3.1-70B-Instruct` is chosen for its accessibility and reproducibility, whereas `xFinder-Qwen` is chosen because it is optimized for accurately identifying and extracting answers from LLM-generated text.

As shown, the accuracies obtained from Llama-classified generations are on par with those of the FTP approach enhanced with prefilling across all datasets and models, indicating that our prefilling strategy is a robust and effective evaluation strategy. The only cases where we observe noticeably lower results compared to open-ended classification are MathQA and ARC-C. The reduced accuracy of both FTP and FTP with prefilling on these datasets suggests that more advanced reasoning is required, and that prefilling alone cannot compensate for this limitation.

A similar trend is observed with xFinder-classified results: in most cases, FTP with prefilling achieves higher accuracies, while for OpenBookQA and Moral Stories, FTP with prefilling and open-ended generation evaluated via xFinder perform roughly equally. For MathQA, results are more mixed, with open-ended generation sometimes providing an advantage, but not consistently across all models. Nonetheless, MathQA results highlight our main message: even when prefilling does not close the

gap to open-ended reasoning in reasoning-heavy tasks, it still provides consistent gains without additional compute.

## C  FULL RESULTS ON FULL-VOCABULARY EVALUATION

Table 6 complements the partial results in Table 2 by reporting full-vocabulary accuracy and First Token Validity Rate (FTVR) for all models and benchmarks. While most models benefit from prefilling, this full breakdown reveals particularly brittle behavior in the Gemma models: both `Gemma-7B` and `Gemma-2-9B` consistently produce 0% full-vocabulary accuracy and 0% FTVR under standard FTP. This failure is due to their tendency to always begin responses with the token 'The', forming preambles like 'The correct answer is' instead of immediately emitting a symbolic option. As a result, their top-1 token never matches the valid label set when decoding is unconstrained. The prefilling strategy successfully overrides this default behavior, enabling proper symbolic alignment and resulting in large performance gains across different MCQA benchmarks.

Table 6: Full-vocabulary first-token evaluation on four MCQA benchmarks for all tested LLMs. The model must predict a valid and correct token as its very first output. Prefilling improves both validity and full-vocabulary accuracy.

| | MMLU | | OBQA | | SIQA | | SCIQ | |
|---|---|---|---|---|---|---|---|---|
| | Acc | FTVR | Acc | FTVR | Acc | FTVR | Acc | FTVR |
| Llama-3.1-8B | 6.4 | 9.5 | 17.8 | 20.4 | 52.7 | 69.6 | 2.2 | 2.4 |
| + prefilling | **64.0** | **99.3** | **80.8** | **99.8** | **71.5** | **99.9** | **96.9** | **100.0** |
| Qwen-2-7B | 61.7 | 85.9 | 80.4 | 93.8 | 71.6 | 90.9 | 94.3 | 95.7 |
| + prefilling | **66.1** | **97.0** | **84.8** | **100.0** | **75.8** | **99.4** | **98.0** | **99.9** |
| Gemma-7B | 0.0 | 0.0 | 0.0 | 0.0 | 0.0 | 0.0 | 0.0 | 0.0 |
| + prefilling | **34.9** | **56.4** | **69.8** | **95.6** | **64.5** | **98.6** | **90.5** | **95.0** |
| Gemma-2-9B | 0.0 | 0.0 | 0.0 | 0.0 | 0.0 | 0.0 | 0.0 | 0.0 |
| + prefilling | **71.4** | **99.2** | **88.6** | **98.6** | **73.6** | **95.3** | **98.2** | **99.8** |
| Zephyr-7B | 38.4 | 65.8 | 56.6 | 76.6 | 53.4 | 80.5 | 20.4 | 22.1 |
| + prefilling | **52.9** | **89.9** | **69.2** | **92.2** | **57.3** | **86.1** | **63.5** | **66.0** |
| Ministral-8B | 27.1 | 37.1 | 77.0 | 89.2 | **71.9** | **94.3** | 9.5 | 9.5 |
| + prefilling | **41.7** | **55.3** | **77.2** | **93.2** | 56.7 | 71.5 | **71.8** | **73.8** |
| Mistral-Nemo-12B | 21.6 | 27.6 | 31.4 | 34.6 | **44.5** | **56.5** | 3.2 | 3.5 |
| + prefilling | **40.7** | **61.9** | **64.8** | **77.2** | 42.8 | 53.9 | **72.0** | **73.1** |
| Phi-4-14B | 8.9 | 10.7 | 36.8 | 42.6 | 30.3 | 35.7 | 10.1 | 11.7 |
| + prefilling | **37.0** | **41.2** | **81.0** | **88.0** | **72.7** | **93.3** | **85.4** | **86.7** |

## D  CALIBRATION METRICS

Here, we provide detailed definitions and explanations of the calibration metrics used in Section 4.5. In particular, Adaptive Calibration Error (ACE) (Nixon et al., 2019) partitions predictions into adaptive ranges such that each range contains an equal number of predictions, mitigating bias from sparsely populated bins and better reflecting calibration across all classes. The adaptive ranges are created by first sorting the predicted probabilities for each class. The sorted list is then divided into $R$ contiguous ranges such that each range contains approximately $\lfloor N/R \rfloor$ predictions, where $N$ is the total number of predictions for that class. This ensures that each range is equally populated, allowing ACE to focus on regions with sufficient data while avoiding the sparsity issues that affect fixed bins. Formally, let $R$ be the number of adaptive ranges and $K$ the number of classes. ACE is defined as

$$\text{ACE} = \frac{1}{KR} \sum_{k=1}^{K} \sum_{r=1}^{R} |\text{acc}(r, k) - \text{conf}(r, k)|, \tag{3}$$

where $\text{acc}(r, k)$ and $\text{conf}(r, k)$ denote the empirical accuracy and average predicted confidence for class $k$ in adaptive range $r$, respectively. Unlike Expected Calibration Error (ECE) (Naeini et al., 2015), which only considers the probability of the predicted class for each example, ACE evaluates calibration for each class separately by including the predicted probabilities for all classes, not just the one with the highest predicted probability.

The Brier score (Glenn et al., 1950) is the mean squared error between the predicted probability $p_i$ for the correct class and the true label $y_i$ across all $n$ examples:

$$\text{Brier Score} = \frac{1}{n} \sum_{i=1}^{n} (p_i - y_i)^2. \tag{4}$$

It quantifies both the accuracy and the calibration of the probabilistic predictions, penalizing confident incorrect predictions.

Log Loss (Hastie et al., 2009; LeCun et al., 2015), also known as negative log likelihood or cross-entropy loss, evaluates the probabilistic predictions by aggregating the log-probabilities assigned to the true classes. For a single example with true class $y$ and predicted probabilities $p_k$, it is defined as

$$\text{Log Loss}(x_i, y_i) = -\sum_{k=1}^{K} y_{i,k} \log(p_{i,k}), \quad (5)$$

where $K$ is the number of classes, $y_{i,k}$ is 1 if class $k$ is the true label and 0 otherwise, and $p_{i,k}$ is the predicted probability assigned to class $k$.

The overall Log Loss across a dataset of $n$ examples is then computed as the average:

$$\text{Log Loss} = \frac{1}{n} \sum_{i=1}^{n} \text{Log Loss}(x_i, y_i). \quad (6)$$

This formulation penalizes predictions that assign low probability to the true class; equivalently, it severely penalizes overconfident misclassifications in which the model assigns high probability to an incorrect class. Compared to the Brier score, which applies a quadratic penalty, Log Loss is harsher on highly confident errors.

## E  LIMITATIONS

This work focuses specifically on evaluating first-token vulnerabilities in MCQA tasks under symbolic decoding setups such as FTP. Our analysis centers on two types of error (*i.e.*, first-token misalignment and misinterpretation) arising from formatting mismatches between model outputs and expected symbolic responses. The scope of our study is also limited to a subset of English-language benchmarks and does not yet consider multilingual or domain-shifted tasks.

While our controlled experiments and prefilling-based mitigation strategy shed light on the fragility of symbolic decoding, we do not examine how other decoding regimes, such as beam search or temperature sampling, might interact with these failure modes. Additionally, our study does not explore whether similar misalignment effects occur in open-ended QA, summarization, or dialogue tasks, where symbolic constraints are looser or absent.

We hope that future work will extend our analysis to broader tasks and languages, and will develop decoding strategies and prompt formats that are inherently robust to structural ambiguity and symbolic misalignment.

## F  LLM USAGE

This work systematically analyzes the performance of LLMs on MCQA tasks. Apart from the experiments described in the paper, LLMs were used only for minor writing polish. They did not contribute to the design of experiments, the analysis of results, or the generation of scientific content.

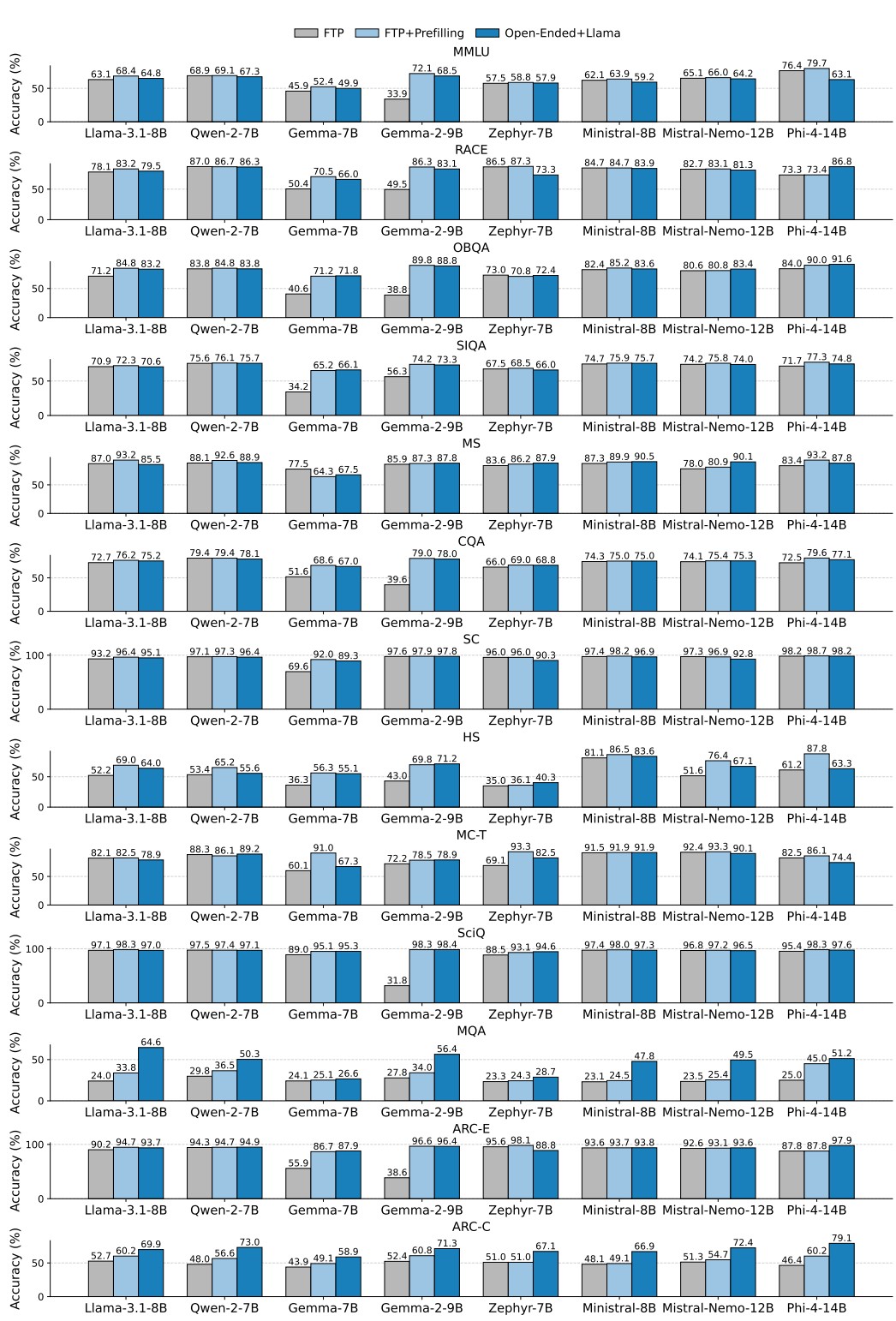

Figure 5: Accuracy comparison on all the benchmarks using standard FTP, FTP with prefilling, and open-ended generation with `Llama-3.1-70B` as classifier. Again, FTP with prefilling consistently outperforms standard FTP and often outperforms or is on par with the more computationally expensive open-ended generation approach.

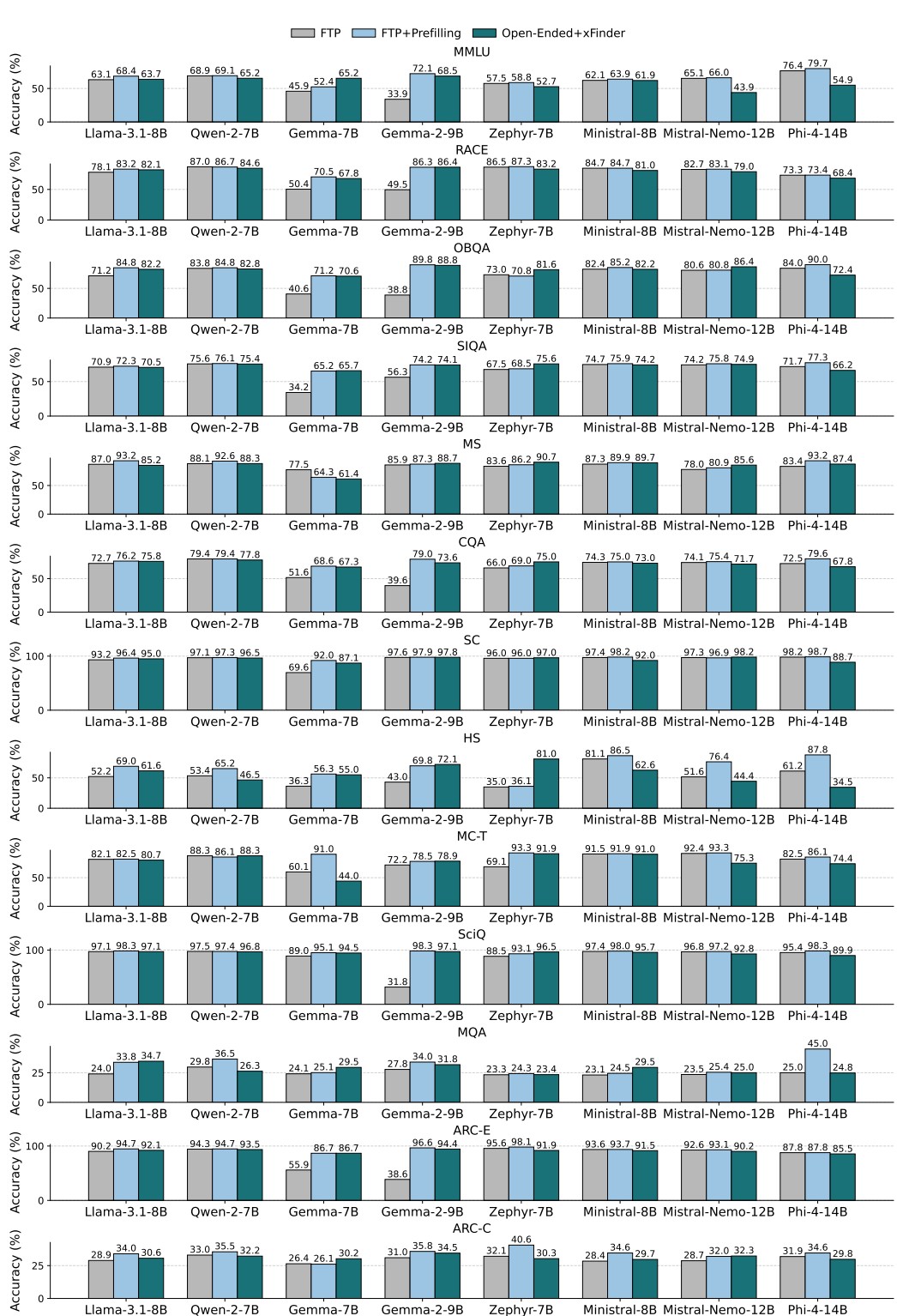

Figure 6: Accuracy comparison on all the benchmarks using standard FTP, FTP with prefilling, and open-ended generation with `xFinder-Qwen` as classifier. Again, FTP with prefilling consistently outperforms standard FTP and often outperforms or is on par with the more computationally expensive open-ended generation approach.

