# OpenReview forum: "Improving LLM First-Token Predictions in Multiple-Choice Question Answering via Output Prefilling"
_ICLR.cc/2026/Conference — ICLR 2026 Conference Withdrawn Submission_

### Official Review · Reviewer_fYb8 · 2025-10-23

**Soundness:** 2
**Presentation:** 3
**Contribution:** 1
**Rating:** 2
**Confidence:** 3

**Summary:**

This paper addresses the issue of generative Language Models failing the output format of MCQA questions even when their output is a generally correct answer. Authors propose to steer models towards giving the final answer (chosen option) without any leading phrases (i.e., "I think that the correct answer is...") by "pre-filling" the end of prompts with structures like ‘The correct option is:’.
They perform extensive experiments applying the pre-filling strategy to numerous LLMs on multiple commonly used MCQA benchmarks, showing noticeable improvement in performance over the baseline solutions. Authors additionally study open-end generation, and perform analysis of the effect that the proposed method causes on the model’s confidence in its answers (calibration), once again showing great improvement over the baseline.

**Strengths:**

- Solid study of prompt engineering effects in MCQA setting. The answer steering is commonly (but implicitly) used in MCQA for quite some time, usually in the form of the line "Answer:" after the list of options in the prompt; however, the necessity and effect of it weren’t properly studied before.

- An exhaustive coverage of multiple commonly used models of various sizes from several families.

**Weaknesses:**

My primary concern with this paper is its somewhat lack of novelty. The proposed method, essentially, is an elaborated prompt engineering for Large Language Models, and the results support commonly known fact that LLMs may be highly sensitive to the prompt format in MCQA. There is no analysis of the models’ innerworking (or training of any new models); and in my opinion this paper doesn't quite correlate with the topic of the conference.
Moreover, the main idea behind the proposed method (steering the model towards directly outputting the option as the first token) was already implicitly implemented in many cases, including the standard format of prompts at commonly used benchmarks like MMLU (where the model is prompted with a question and a list of options, followed by "Answer:" after which the model is required to autofill the answer option, as described in the original publication for MMLU). The paper under review suggests replacing in this prompt "Answer:" with a more sophisticated construction, but does not change the principle behind it.

Finally, from the experimental setup explanation in the paper, it seems that First Token baseline was run with the prompt containing only the question and list of options (without finishing parts like "Answer:" for MMLU), which deviates from the commonly applied setup  and, potentially, worsens baselines performance (especially in the case of token selection from the full dictionary in section 4.4).

**Questions:**

- Could you please provide prompt template for baseline methods?
- Do you have any insight on what effect the prefilling technique has on non-instruct tuned (a.k.a. "base") models?
- Could you please analyze the influence of the "technical" tokens (|<start_header_id>|, <|eot_id|>, etc.)? In my experience of using LLaMA models locally via the Transformers library, inclusion of the role-specifying parts in the prompt (e.g., "<|eot_id|><|start_header_id|>assistant<|end_header_id|>") actually degraded the performance of the model on MCQA tasks (and I've hardly ever used such commands before).
- Do you have any insight on how prompt prefilling affects the model's performance in the case of the few-shot prompting?

---

### Official Review · Reviewer_rSt9 · 2025-10-27

**Soundness:** 2
**Presentation:** 3
**Contribution:** 2
**Rating:** 6
**Confidence:** 3

**Summary:**

This paper investigates the fragility of FTP evaluation in multiple-choice question answering tasks and proposes output prefilling as a simple yet effective solution. The authors identify two key failure modes in standard FTP: first-token misalignment, where models generate tokens outside the valid answer set, and first-token misinterpretation, where a valid token appears grammatically but doesn't represent the intended answer. By repurposing the prefilling attack from AI safety research, they prepend structured natural-language prefixes to the model's output, mechanically steering the first generated token toward valid symbolic answers.  The method achieves performance comparable to computationally expensive open-ended generation approaches while requiring no model modification or additional inference cost. The main contribution is a comprehensive empirical validation showing that this zero-cost intervention significantly enhances the reliability of FTP-based MCQA evaluation.

**Strengths:**

### 1. **Simple and Practical Solution**

The proposed output prefilling method is remarkably simple to implement, requiring no model fine-tuning, architecture modification, or additional computational cost. By prepending a natural language prefix to guide the first token generation, the method achieves substantial improvements across diverse models and benchmarks while maintaining zero inference overhead. This practical design makes it immediately deployable in real-world evaluation pipelines.

### 2. **Comprehensive Experimental Validation**

The paper provides extensive empirical evidence across 8 language models and 14 benchmark datasets, demonstrating consistent improvements in accuracy, calibration quality, and output consistency. The evaluation is multi-faceted, including full-vocabulary analysis, calibration metrics, comparisons with open-ended generation, and ablation studies with 10 template variants. This thorough experimental design substantially strengthens the credibility of the proposed approach.

### 3. **Clear Problem Identification and Analysis**

The authors systematically identify and categorize two fundamental failure modes of First-Token Probability evaluation: first-token misalignment and first-token misinterpretation. The introduction of new metrics such as Coverage Degree and First-Token Validity Rate provides quantitative tools for diagnosing these issues. This clear problem formulation and diagnostic framework contribute valuable insights to the community beyond the specific technical solution.

**Weaknesses:**

### **Insufficient Analysis of Extreme Performance Variations**
While the paper reports dramatic improvements for models like Gemma-2-9B (33.9% → 72.1% on MMLU), the explanation provided in Appendix C attributes this to the model's tendency to begin with "The" rather than symbolic options. However, this raises deeper questions: Why do Gemma models exhibit this behavior while others don't? Is this a tokenization issue, an alignment artifact, or a fundamental architectural difference? A more thorough investigation of these model-specific behaviors would strengthen the paper's contribution to understanding LLM evaluation challenges.

**Questions:**

Can you show how prefilling mechanistically works? E.g., token probability shifts or attention changes.When and why does prefilling fail? What predicts its effectiveness across tasks?

---

### Official Review · Reviewer_fSDS · 2025-10-27

**Soundness:** 2
**Presentation:** 2
**Contribution:** 1
**Rating:** 0
**Confidence:** 5

**Summary:**

The paper proposes inserting a short natural-language prefix ("Given the question and the possible options, my answer is:") immediately after the assistant turn to boost LLM performance on Multiple-Choice Question Answering (MCQA). Across several models and MCQA benchmarks, this formatting tweak improves both accuracy and calibration. The technique is called "output prefilling". The authors further compare prefilling to a "prompt-side" baseline (“Please answer only with A/B/C/D”) and to open-ended generation evaluated via external LLMs.

**Strengths:**

The idea of the paper is easy to understand and reproduce.

**Weaknesses:**

- The entire paper revolves around adding a single hard-coded natural-language prefix before the model's first token. This is a trivial form of prompt engineering, yet the paper frames it as a novel methodological advance. There is no conceptual depth or new modeling insight - only an empirical observation that some phrasing slightly improves results on MCQA benchmarks. This is far below the threshold of novelty required for A* conference.
- No systematic ablations of prefix content are provided. Appendix A briefly mentions testing 10 different templates but only reports average accuracy and standard deviation across them, without identifying which ones perform best or exploring the connection between their linguistic properties (length, directness, formality, etc.) and the results.
- Although calibration metrics are reported, there is no qualitative analysis of why they improve. Prefilling might simply reduce entropy by forcing deterministic symbolic outputs - a trivial statistical effect, not genuine improvement in uncertainty modeling.

**Questions:**

No questions

---

### Official Review · Reviewer_iGYp · 2025-10-31

**Soundness:** 4
**Presentation:** 4
**Contribution:** 1
**Rating:** 2
**Confidence:** 5

**Summary:**

The authors propose "output prefilling", a simple technique designed to enhance LLM accuracy in MCQ benchmarking. By appending a phrase such as "The correct option is: " to the prompt (importantly: in a chat scenario, the phrase is appended to what the assistant is generating, not to the user query or the system prompt), accuracy (and other metrics such as FTVR) can be improved.

This is a very, very well-written paper that thoroughly investigates a very, very simple idea.

**Strengths:**

The authors convincingly demonstrate, through a variety of experiments, that their output-prefilling technique can improve MCQ benchmarking performance on a handful of models.

The visualizations are tidy and interpretable.  The paper flows nicely.  References seem to be thorough and accurate.

When it works, the method can

**Weaknesses:**

While the empirical work is solid, the authors do a comparatively poor job of motivating the significance of this result.  I do not question that they are the first to do this specific thing, but I wonder if these results will matter "in the real world".

This is perhaps evidenced by the lack of a "Limitations" discussion in the paper.  As I see it, the limitations -- both to the method, and to the evaluation -- are somewhat severe:

* Is this method helpful with reasoning models?  It's unclear.  I don't doubt that you could append the key phrase after the reasoning tokens are generated and as the agent is beginning to generate output tokens, but would it have the same effect?  I doubt it, because I expect that somewhere in reasoning the agent would have decided/noticed that it should limit its output to a single letter. (This would be especially true with the prompt-engineering variation of the task).

* This leads me to wonder: why weren't any reasoning models tested?

* These are all relatively old models. Most were released sometime time in 2024.  Why weren't more recent models tested?  For example, you test the Qwen2 series, but not the Qwen3 series; you test the Llama 3.1 series, but not 3.2 or 3.3 or 4.0; many of the best OSS models aren't tested (Deepseek, Kimi, etc.).

* These are all relatively small models. While I appreciate that research groups are often compute-bound, it seems like you could have tested a few 70-80b models without too much difficulty.  I might expect to see larger models perform very differently, as they are typically better at instruction following.

So: is this work relevant to reasoning models, or flagship models, or large-parameter models? Or is this just a simple way to take bad models and make them better?

**Questions:**

See weaknesses section.

---

### Official Review · Reviewer_SX3G · 2025-11-03

**Soundness:** 2
**Presentation:** 3
**Contribution:** 1
**Rating:** 2
**Confidence:** 4

**Summary:**

This paper addresses a known fragility in multiple-choice question answering (MCQA) evaluation with large language models: the First-Token Probability (FTP) method can fail when the model’s first generated token doesn’t align with a valid answer option.

To solve this, the paper proposes “output prefilling”: inserting a fixed natural-language prefix (e.g. “The correct option is:”) at the beginning of the model’s output. By prepending this template to the model’s response (without changing the model’s weights), the method aims to bias the model’s first output token to cleanly be one of the provided options.

Through extensive experiments on several MCQA benchmarks (including MMLU, OpenBookQA, Social IQa, SciQ, RACE, ARC, etc.) and across a range of open LLMs (Gemma-2B/7B/9B, Zephyr-7B, Llama-3.1-8B, Qwen-7B, Mistral-Nemo-12B, Phi-4-14B, etc.), the authors demonstrate that prefilling dramatically improves evaluation results. The method is a simple formatting hack that can be applied at inference time.

**Strengths:**

1) The paper tackles a concrete problem (fragility of first-token multiple-choice evaluation) and presents a simple yet effective solution. The output prefilling strategy yields consistent and substantial accuracy improvements across a wide range of models and benchmarks.

2) Strong Empirical Evidence and Rigor: This work provides extensive experimental validation.

3) A major strength is that the proposed solution is extremely simple to implement and computationally cheap.

4) The paper is clearly written and organized

**Weaknesses:**

1) The paper has limited technical novelty. The core technique – inserting a fixed prefix to guide the model’s output – is not a fundamentally novel algorithmic contribution. It is an adaptation of a known prompt injection tactic from the AI safety literature (the “prefilling attack”). The experiments largely reiterate a well‑established point: multiple‑choice QA performance is highly sensitive to prompt formatting. The paper neither investigates model internals nor trains new models.

The core mechanism—biasing the model to emit the option label as its very first token—has long been used implicitly. Standard protocols such as MMLU prompt with a question and options, then append “Answer:” so the model autocompletes the option label, as outlined in the original MMLU paper. This submission replaces that approach with a more detailed template but does not alter the underlying principle.

The authors’ application of this idea to MCQA is novel, but some may view the work as a relatively straightforward trick rather than a deep innovation. In essence, the paper’s value is in recognizing and fixing an evaluation issue more than in inventing new modeling techniques.

2) The experiments, while extensive for MCQA, are limited to multiple-choice QA tasks in English. The authors specifically note they did not explore other task formats (like open-ended QA, dialogue, or generation tasks) where similar misalignment issues could occur. It’s unclear how broadly the prefilling approach generalizes. For example, would this technique help if the model needed to output a structured list or a specific format in other contexts? The paper doesn’t investigate tasks beyond QA, nor does it cover multilingual benchmarks. Thus, the contribution, as strong as it is for MCQA, has a somewhat narrow application domain in its current form. The significance for other settings remains an open question.

3) While results are overwhelmingly positive, the paper does not deeply analyze scenarios where prefilling might not help or could possibly hurt. For instance, in Table 2, one model (Mistral-12B on Social IQa) saw a slight decrease in accuracy with prefilling (44.5% to 42.8%, if I read correctly) despite an increase on others. The authors do not comment on this anomaly – is it within error margin or is there a specific reason prefilling didn’t help that case? More broadly, one wonders: are there any situations where prefilling could bias the model towards an incorrect choice or interfere with the reasoning process? The paper’s analysis did not highlight any clear failure cases or trade-offs (which is good news, but also surprising). A bit more discussion on when/why the method might sometimes underperform or any observed downsides (even trivial, like slightly longer prompts) would have strengthened the rigor. As it stands, the technique appears almost too universally positive, which warrants careful consideration.

4) The paper could be criticized for missing a couple of baseline variants. They compare against one prompt engineering instruction (“answer only with [A/B/C/D]”), but there are other ways one might try to enforce format without output injection – for example, explicitly instructing the model in the user prompt not to provide explanation and just give the option letter. It’s implied their prompt-side baseline covers this, but some detail on how that prompt was phrased would help (the paper only briefly mentions appending “Please answer only with [OPTIONS LIST]”). Also, when comparing to open-ended generation, they use external classifiers. Another reasonable baseline could have been simple regex extraction of the option from the model’s free-form answer (many practical evals do this).

5) The study fixes the prefilling template to one default phrase (“Given the question and the possible options, my answer is: ”). It’s not reported whether this exact wording was optimized or if other variations were tried. The effectiveness might depend on phrasing nuances or model tuning (some models might respond better to different wording). A potential weakness is the lack of exploration into prompt sensitivity: if a user naively chooses a less optimal prefix, would the improvements diminish? The paper doesn’t explore different templates or provide guidance on choosing a good prefix beyond the one example. This could limit practical adoption, although the chosen template seems general and worked well across tasks. It would strengthen the work to know that prefilling is robust to wording or to have justified why the chosen phrase was used (e.g., was it inspired by the AI safety literature examples?).

**Questions:**

1) While the paper focuses on multiple-choice QA, do the authors believe the output prefilling strategy could help in other scenarios? For instance, could a similar approach improve truthfulness or format in open-ended QA (by prefilling an answer prefix), or help structure outputs in tasks like summarization or code generation?

2) How sensitive are the results to the exact phrasing of the prefilling prefix? The paper uses one default template across all experiments. Did the authors try alternative wordings (even minor variations like “The correct answer is:” vs “The correct option is:” etc.) and observe any differences?

3) Could the authors shed light on why prefilling works so effectively from the model behavior perspective? The paper mentions exploiting the model’s “normal cognitive biases” via the structured prefix. Do the authors hypothesize that the prefix puts the model into a more “answer-oriented” mode (perhaps due to fine-tuning on Q&A format data where answers often start with phrases like “The answer is”)?

4) The study mostly evaluates open-source models where one has full control over token-level output. How would one implement output prefilling in a black-box API setting (like OpenAI’s ChatGPT)? Is it as simple as including the prefix in the system or user prompt?

5) The results show a notable improvement in calibration with prefilling. Can the authors clarify how calibration was measured (ECE, log loss, or simply looking at confidence vs accuracy curves)? And why might prefilling improve calibration “by a substantial margin”? Is it because the model’s probability mass is re-distributed more meaningfully when it is confident about a structured answer format (versus possibly predicting some irrelevant token with high probability)?

---

### Note · Authors · 2025-11-20

I have read and agree with the venue's withdrawal policy on behalf of myself and my co-authors.